# Recognizing Determinants to Smallholders' Market Orientation and Marketing Arrangements: Building on a Case of Dairy Farming in Rural Kenya

James Wangu [1,*] , Ellen Mangnus [2] and A. C. M. (Guus) van Westen [1]

1   Department of Human Geography and Spatial Planning, Utrecht University, Vening Meineszgebouw A Princetonlaan, 3584 CB Utrecht, The Netherlands; A.C.M.vanWesten@uu.nl
2   Department of Social Sciences, Wageningen University & Research, 6700 HB Wageningen, The Netherlands; ellen.mangnus@wur.nl
*   Correspondence: j.m.wangu@uu.nl or jameswangu@gmail.com

**Abstract:** Smallholder commercialization is central to international development policy and practice. As a result, several arrangements to foster market linkages are being implemented. Especially popular are farmers' organizations, which are believed to be owned, controlled, and financed by smallholders. As such, their design is considered inclusive given every household in a community is theoretically allowed to become a member, and the governance and management structure encourage participatory decision-making. However, even in the context in which farmers' organizations are actively promoted, a notable proportion of smallholders may not be able to engage in market-oriented production or may opt for the existing alternative marketing arrangements, as dictated by individual households' socioeconomic characteristics. Focusing on the case of smallholder farming in Olenguruone, Nakuru county, Kenya, where a donor funded dairy farmers' cooperative marketing arrangement is promoted alongside existing marketing opportunities, the present research investigated the factors that determine smallholders' commercial farming orientation and marketing arrangements. It employed a case study approach, combining both quantitative and qualitative research methods for a more complete empirical inquiry. The findings demonstrate that irrespective of the external support provided through marketing opportunities such as farmer organizations, smallholders' engagement in commercial farming and marketing is dictated by the socioeconomic attributes and market perceptions that are heterogeneous among households in a smallholder community.

**Keywords:** heterogeneity; farmers' organizations; cooperative; agribusiness; inclusion; donor



## 1. Introduction

Poverty and incidences of hunger and malnutrition remain high among smallholders in Sub-Saharan Africa, yet smallholder agriculture is considered the key solution to these challenges. Smallholders constitute farmers who rely on small farms (family farms) and are likely to experience marginalization in terms of access to land resources, input, technology, and information available to farmers with larger farms [1,2]. Improving their small farms' productivity and income via increased market orientation (commercialization) has for a long period been among the central focus of the international development policy and practice [3,4]. Linking them to the market, often portrayed as being imperfect, remain on top of this agenda [5,6]. Indeed, poor infrastructure, failing institutions, lack of information, and high transaction costs, among other challenges, do prevent smallholders from capturing the benefits of the growing domestic demand for food and a rapidly globalizing agricultural economy [6,7].

Against this background, there are numerous interventions geared toward improving smallholders' access to produce markets. However, while agricultural commercialization

does present opportunities for smallholders to enhance their livelihoods, rarely do scholars and policymakers question whether every farming household can effectively participate or benefit from their involvement, even with the external support. For the externally supported marketing opportunities to benefit smallholders, the support must adequately address the farming and marketing needs of the smallholders. This paper explores smallholders' market orientation and inclination towards specific marketing arrangements. It concentrates on establishing the determining factors to the latter aspect. Doing so will provide a better understanding of the potential success of donor funded interventions promoting specific marketing arrangements: an issue seldomly discussed in the literature.

In studying smallholders' market orientation and marketing strategy, we apply the actor-oriented approach, which focuses on how actors' (in this case smallholders') attribute affect their choices [8]. According to Long [9], 'rural development represents a complex drama about human needs and desires, organizing capabilities, power relations, skills and knowledge [ . . . ]'. The actor-oriented approach, therefore, is a useful lens for '[ . . . ] explaining differential responses to similar structural circumstances, even if the conditions appear relatively homogeneous' Long [10]. In development interventions smallholders tend to be perceived as a passive homogenous group whose solution can be addressed by a specific intervention. However, as Long [9] adds, smallholders should not be seen as 'passive recipients of intervention, but as active participants who process information and strategize in their dealings with various local actors as well as with outside institutions and personnel'. He (ibid) stresses on a need to put actors' choices in context which includes, but is not limited to, the distribution of power and/or resources, cultural disposition, past experiences, individual concerns, personal habits, lifestyle, peculiarities, emotions, and feelings [11]. In the present research we limit the context to individual smallholders' social and economic attributes, past marketing experiences, and individual concerns.

Under today's most popular pro-poor approach—inclusive business [12,13]—several arrangements to link smallholders to national, regional, and global markets have been pursued. Examples are cooperatives, contract farming, outsourcing, and market linkage through a lead farmer, but also the establishment of rural markets and auctions [14–16]. Especially common are farmer organizations such as cooperatives [6,14,17]. Farmer organizations have become the centerpiece of pro-poor market development interventions in Sub-Saharan Africa [6,18] and are presumed to embrace the ideology of bottom-up empowerment. Owned, controlled, and financed by smallholders, their design is considered inclusive given that every household in a community is theoretically allowed to become a member, and the governance and management structure encourage participatory decision-making [19].

The success of a farmer organization is pegged on its ability to achieve economies of scale, both in production and marketing through collective action [17]. To benefit from the collective opportunity, a smallholder must be a member of an organization that qualifies in terms of its capacity to enhance local production and support access to the market. However, numerous studies have shown that participation in a farmers' organization is determined by a wide range of factors, including production capacity as dictated by productive resources, education level, age, information, social capital, and distance to the produce selling center, among others [20–22]. These findings suggest that not every smallholder gets to be part of a farmers' organization. At the same time, a few studies have demonstrated that smallholders' marketing strategies, i.e., whether to join a farmers' organization or use other marketing options, are similarly influenced by various socioeconomic characteristics [23–25]. If this applies in the context where farmers' organizations are actively promoted through donor or government funded interventions, it would imply there is a discrepancy between the promoted marketing arrangements and the smallholders' needs. Given donor persistence in farmers' organizations, understanding the local outcome is critical for the future interventions.

The premise of this paper is that smallholders are diverse in terms of their socioeconomic attributes, farming experiences, and concerns, which in turn dictates participation

in commercial farming, and individual households' goals and attitudes toward specific marketing strategies. By goals we mean what the farmers aim to achieve through a specific marketing arrangement, and by attitudes we mean marketing preferences as determined by experiences and factors unique to individual households. It is widely upheld that farmers' goals and attitudes vary at an individual level [26–29]. Building on the literature on farmers' technology adoption, heterogeneity arising from, among other factors, farmer characteristics, asset endowments, risk preferences, and intertemporal factors, determines what farmers grow, their use of technology, and their land management practice [28]. For instance, it has been established that farmers' income, resource endowment, and/or access to credit affect crops choice, farming systems, and inclination towards a new crop, systems, or technologies [30,31].

Building on this background, the present research focus on a case of a donor funded farmers' cooperative marketing arrangement that is promoted alongside other existing marketing opportunities in rural Kenya. To better grasp issues in smallholder dairy farming and marketing landscape in the research community, it is imperative we delve into the evolution of Kenya's dairy sector, its relevance to the local economy, and current challenges.

*Kenya's Dairy Industry*

The year 1925 can be seen as the naissance of Kenya's dairy industry, following the establishment of the Kenya Co-operative Creameries (KCC) by European colonialists with the aim of advancing an agenda on milk production, processing, and marketing in the country [32,33]. When they left before independence (1963), their cattle were offered to the local people [34] and land was 'subdivided and redistributed in line with the land reform movement' [35], allowing smallholders to become the majority of milk producers in the country [36]. This necessitated KCC to incorporate smallholders in the enterprise, which it did by establishing the necessary infrastructure, most importantly regional cooling plants [35]. For decades the dairy sector was a monopoly under KCC, a state-led entity controlling the entire chain: milk collection, processing, and marketing [35,37]. This arrangement provided for the government to dictate the prices at the producer and consumer levels [37]. However, due to the entry of new cooperatives as dairy processors: Meru Central Farmers Co-operatives Union (MCFCU) in 1983 and the Kitinda Dairy Farmers Co-operatives Society (KDFCS) in 1986, KCC lost its monopoly privilege [35]. Since then, Kenya has seen a proliferation of dairy companies and cooperatives that have entered and exited the market over time.

The dairy sector in African countries has seen a notable support from the international donor and development community. Between 1975–85, donor funding to the sector in developing countries amounted to USD 80 million a year, 25% of which came from local governments [38]. In Kenya, these funds enabled the government to provide subsidized dairy production services, including breeding, animal health, and milk production [33,35]. This sector approach, however, ended following the government's adoption of a new policy on Economic Management for Renewed Growth that 'emphasized the need for small enterprises to be nurtured as beacons for future growth' [39]. This structural change that called for 'price decontrols, liberalization of marketing, government budget rationalization, and parastatal reform' [35], led to a shift from the public to the private sector as an essential services provider but resulted in failure [40]. Furthermore, the implementation of these radical economic reforms, in addition to political influences and corruption within KCC led to the company's dissolution [32,35]. As a result, informal diary traders, alongside a few private processors, proliferated to meet the processing and marketing demand previously controlled by KCC [34,35].

Kenya's dairy industry is regarded as vibrant [41]. According to the country's Dairy Board (KDB), the dairy sector is Kenya's largest agricultural sub-sector and a significant contributor to the national economy, accounting for '14% of the agricultural GDP and 4% of the national GDP' [42]. The growth of the sector is notable, experiencing an annual average rate of 5.3, 7, and 5.8 litres in milk production, processing capacity, and per

capita consumption, respectively [42]. At an annual average of 117 L per year, Kenya's per capita milk consumption is among the highest in Africa [43]. Many Kenyans derive their livelihood from dairy farming; an estimated 1.3 million households keep dairy cattle [44]. Smallholder production accounts for more than 80 percent of the total raw milk output in Kenya, with the rest coming from large scale producers [45]. However, it is important to note that while a proportion of households engage in commercial dairy farming smallholders' communities in the country, there are also many that are not involved. Hence, they cannot directly benefit from the growing industry.

The raw milk market is dominated by informal traders, who control up to 80% of the total product of which only 16% undergoes either home or artisanal processing [46–48]. Middlemen, commonly known as brokers and often viewed negatively as being exploitative to farmers, are among the key informal traders [48]. The rest is processed and marketed through competitive formal channels—those that do some processing prior to sale. While the number of the registered dairy processors in the country exceeds 30, only a few dominate the market. According to the available data, up to 80% of milk is traded through formal channels, including Brookside, New KCC, and SnipKnit [46,48]. Together with other relatively small but reputable players these three big players are accused of operating in an oligopolistic manner [41,46,48].

Cooperatives and farmers' groups are said to be crucial players in milk production and marketing in Kenya and are increasingly encouraged to be formed [40,41,46,49,50]. With respect to production, they provide various forms of support to their members, including supply of subsidized inputs, animal health services (artificial insemination and veterinary services), access to credit facilities, extension services, and bargaining power. As marketing agents, they collect, bulk, cool, and distribute milk either after processing, or sell it directly to processors and/or consumers. The bulking service is recognized by the government as essential for lowering the marketing costs for the farmers, thereby contributing to better returns [40]. In total, there are about 334 cooperatives and farmers' groups spread across the country according to the recent data from KDB [51].

Despite the positive outlook, Kenya's dairy industry is faced with numerous challenges. Firstly, the milk production is considerably low due to, among other factors, shrinking farm sizes, poor animal husbandry, low quality feeds, diseases, a declining genetic base, and the impact of climate change [40,50]. Secondly, although there is a wide range of milk markets, the key players are faced with many challenges, including costly milk processing, an unstable supply of milk—characterized by cycles of scarcity and abundance—milk quality and safety issues, poor infrastructure, and risk of unhealthy competition from growing oligopolists in milk processing: problems linked to poor policy, regulation, and enforcement in the country [40,41,43,45]. Thirdly, because of the oligopolistic nature of the milk market, sellers are forced to take the price offered by the big processors [41,52]. These milk market's challenges keep producer prices down to a point where most farmers' revenues go below the cost of production [41]. Poor market access and low milk prices deter farmers from increasing their investment in milk production, thus these challenges constitute the key hurdles to the sector's growth [40,49].

Against these backdrops, the Kenyan government and the international development community alike perceive the dairy sector as a high potential agricultural sub-sector to boost local livelihood and food security [40,53–55]. To do so, various interventions aim at addressing the sector's growth bottlenecks by improving dairy productivity and marketing. The former effort entails increasing the production capacity of smallholders, whilst the latter's focus is on marketing formalization. It is argued that the heavy involvement of the informal traders makes quality control and standard enforcement difficult [40]. It is conceived that farmers' organizations, particularly cooperatives, present the best pathway for the smallholders to address these issues.

Recognizing that cooperative marketing arrangement in Kenya's dairy industry is conceived as among the ideal options for smallholder commercialization by government and the donor community, it is also important to presume that other existing marketing

opportunity serves a significant role in addressing farmers marketing needs not met by cooperatives. Given that little is known about the key determinants to smallholders' market orientation and inclination towards specific marketing arrangements, it becomes a relevant area of research. Hence, the aim of the present research. The study findings will be crucial for continued efforts to improve ways for advancing farmers' commercialization where applicable, as means to enhance their livelihoods and food security situations.

## 2. Materials and Methods

### 2.1. Case and Study Area Description

This paper is based on fieldwork carried out in Olenguruone, a rural smallholder community located in the lower western part of Nakuru county, Kenya. Bordering the Mau Forest, Olenguruone has a favorable climate for tea and livestock production. Due to regional cold and wet annual weather conditions, food crops do not do well. Hence, there are incidences of food and nutrition insecurity. Through increased promotion by the local government and donor organizations, commercial dairy farming has become an essential contributor to local livelihoods. The practice is gaining popularity locally as it is considered more profitable than traditional livestock keeping. Development campaigns that encourage commercial dairy farming as a high potential undertaking to enhance local livelihood and food security amid diminishing farm plot sizes have been central to the growing interest. Donor organizations such as Denmark's Development Cooperation (DANIDA) and the SNV Netherlands Development Organization (SNV) have been promoting the practice through free farmers' training and by linking farmers to formal markets. Besides the external campaigns, members of the local elite who have attained some success in commercial dairy farming also serve as a source of influence in the community.

Currently, the dairy farmers in Olenguruone have three marketing options. One, a cooperative: Olenguruone Dairy Farmers Cooperative Society (ODFCS), which was established through the support of DANIDA. The cooperative acts as the marketing link between the farmers and a private company: *Happy Cow*, a national dairy manufacturing private company that supplies dairy products to leading supermarkets, restaurants, and hotels in the country [56]. In the period that this research took place, this arrangement was facilitated by a public–private partnership: Kenya Market-led Dairy Programme II (KMDP-II). KMDP-II, implemented by SNV and partners between 2016–2019, was financially supported by the Embassy of the Netherlands in Nairobi [53]. The goal of the program was to increase the competitiveness of the Kenyan dairy sector by enhancing smallholder access to inputs, training, and extension services, and formal market, improving sector management and governance, international linkages and partnership, milk quality, and providing policy and sector support [57,58]. The second market player in Olenguruone is Brookside, regarded as the largest milk processing company in Kenya [59,60]. The third market player consists of brokers, a collection of mobile individuals. Often using motorcycles, they buy milk at the smallholders' farm gate and supply to members of the community not producing their own milk, local restaurants, hotels, schools as well as to ODFCS and Brookside.

### 2.2. Data Collection

The present research applied a case study approach. It refers to 'an empirical inquiry that investigates a contemporary phenomenon within its real-life context; when boundaries and context are not clearly evident; and in which multiple sources of evidence are used' [61]. Mixed methods is a methodology of research that consistently combines quantitative and qualitative approaches, and is known to be embraced in case study research, was employed [62,63]. A total of 333 individuals participated in this study. A cross-sectional survey involving 300 households was conducted between December 2018 and February 2019. The survey questions were designed to capture households' key social and economic characteristics, dairy farming practices for households involved in commercial dairy, marketing choices, and motivations. It encompassed four samples, three of which were

generated based on the marketing outlets where smallholders in Olenguruone sell their milk. The survey respondents included: non-dairy farmers (N = 56), dairy farmers selling milk to ODFCS (N = 154), dairy farmers selling milk to Brookside (N = 49), and dairy farmers selling milk to brokers (N = 42). Non-dairy means farmers not engaged in milk production for sale, either because they do not keep dairy cattle, or their cattle do not produce enough to allow for commercial practice. Non-dairy farmers and dairy farmers selling milk to Brookside and brokers, were sampled through the snowball technique with the help of randomly surveyed ODFCS members. Most of the farmers in Olenguruone worked with the cooperative, hence the higher number of those recruited for this survey.

Qualitative data were gathered through semi-structured interviews and focus group discussions to triangulate and validate the findings from the quantitative (survey) data. The interviews were carried out with seven key informants to better understand the scope of commercial dairy farming in the community. Five of them were milk transporters (Transporter 1–5) for the cooperative. The other two were the cooperative's extension officers, in charge of milk production and dairy husbandry (Extension 1), and production improvement (Extension 2). In addition to key informants, five farmers (one respondent) per group were also interviewed. Three focus groups discussions were made up of 6–8 farmers—men and women. The three groups represented farmers selling to the cooperatives and brokers, and those not engaging in commercial dairy farming. Brookside farmers were not involved in the focus group discussions due to unavailability. The interviews and focus group discussions were complemented by observations and informal talks with smallholders and cooperative staff during the field visits.

### 2.3. Data Analyses

Social and economic attributes affect any farmer's farming practices [64]. We set out to explore whether smallholder households' agricultural production assets, education level, age, and non-farming income influence their participation in commercial dairy farming. For inter-household economic data consistency and reliability we limited the production assets variables to: farm plot sizes, crops, and livestock [29]. Since livestock species vary in size or value, to ensure a standardized unit of measurement as a means of inter-household comparison, this paper adopted the Tropical Livestock Unit (TLU) measurement tool. TLU is a weight-based species exchange ratio used to compare tropical livestock [65,66]. Through TLU '[ ... ] different species of different average size can be described by a common unit and compared' [66]. We use the TLU conversion factors (cattle = 0.7, sheep and goats = 0.1, and chicken = 0.01) as presented by Jahnke [67].

For those leaning to the market production, given a range of local marketing opportunities, they decide based on their needs where to sell their milk. By needs we refer to the external support to improve production such as input convenient access and subsidies, credit, and extension services, amongst others, that are common in cooperative and contract farming business arrangements. A smallholder in need of production support as dictated by their production assets is likely to work with a marketing outlet that provides such support. Furthermore, as Berkhout et al. [29] indicate, 'additional income allows farmers to move away from subsistence production and allows for a more market-oriented production strategy'. Market preference is also influenced by the market features such as prices, payment period, and supply chain transaction costs [28]. For instance, higher prices and shorter payment periods are likely to motivate some smallholders' marketing choices.

Of the social aspects, education level and age are known to influence smallholders' cooperative membership [68–70]. According to Bernard and Spielman [70], a higher education level increases the chance of a cooperative membership. Fischer and Qaim [69] find that young people are less interested in joining collective marketing. We also explore smallholders' individual marketing experiences in the past and current concerns on milk marketing, particularly due to widely known continuous failure of cooperatives in Kenya, and the resentment at middlemen for lack of consistency and reliability.

The quantitative data were analyzed using STATA (version 13). The descriptive outputs derived from these data included frequency and summary tables, and *t*-tests. These outputs were used to describe the households' farming characteristics and to illustrate the households' heterogeneity. To assess smallholders' diversity influence on marketing choice, we conducted a multinomial logit model that allowed us to estimate the probability of marketing choice ('0' = Brookside, '1' cooperative (ODFCS), '2' = brokers) based on selective socioeconomic attributes. A multinomial logit regression 'is used to predicting categorical placement' in instances where 'more than two dependent variables' are involved—(smallholder marketing choice), 'based on multiple dependent variables'—(households farming characteristics)' (Starkweather and Moske [71]. The statistical results were triangulated by the qualitative data, which were analyzed for themes, patterns, and quotes.

## 3. Findings

### 3.1. Olenguruone Smallholder Households' Characteristics

Typical of rural smallholder communities in developing countries, nearly all the households surveyed practice mixed agriculture where they keep at least one kind of livestock besides crops production. At least 96% of the households owned cattle and 68% owned a goat. Alongside the livestock, maize, tea, and potato are important sources of local livelihood, and are produced by 90%, 55%, and 53% of the households, respectively.

Table 1 presents the findings on the households' characteristics in the four respective groups. It appears that the farmers in the different groups vary in terms of personal and farm characteristics. Among the notable factors is the fact that farmers in the non-dairy group own the smallest farm plot sizes and the least livestock in the community and have the lowest annual income. The results show that farmers in the cooperative category, and particularly those in the Brookside group, have relatively more farming assets: farm plot sizes and livestock. Overall, the marketing options appear aligned with the farmers' socioeconomic features, whether they are engaging in commercial dairy farming or not.

**Table 1.** Comparing smallholder households' characteristics.

| | Marketing Options | | | |
|---|---|---|---|---|
| **Variables** | **Non-Dairy (Mean)** | **ODFCS (Mean)** | **Brookside (Mean)** | **Brokers (Mean)** |
| Age | 43.7 | 49.3 | 48.1 | 44.0 |
| Family size | 3.9 | 4.9 | 3.8 | 3.6 |
| Farm plot size (acres) | 3.3 | 6.4 | 6.6 | 4.0 |
| Total annual income (KES) | 239,802.5 | 301,430.5 | 344,298.3 | 262,711.4 |
| TLU | 3.0 | 5.7 | 5.8 | 4.0 |
| Dairy TLU | 0.6 | 1.5 | 1.7 | 1.2 |

Figure 1 provides information on the influence of key household characteristics to engaging in commercial dairy farming and the choice of marketing arrangement (%). The households' gender representation in all the groups seems to be relatively similar, with the majority of the households (%) being male headed. To a certain extent, the household's head level of education, specifically secondary education, seems to a primary factor to whether the household engages in commercial dairy farming or not and marketing channel. It appears that households not engaging in commercial dairy farming are more likely to pursue businesses and/or wage labor. Only a few of the farmers in all groups seems to be formally employed. Access to loans seems to be aligned with different marketing options. The majority of the farmers in Olenguruone appear to have access to an information medium (television/radio).

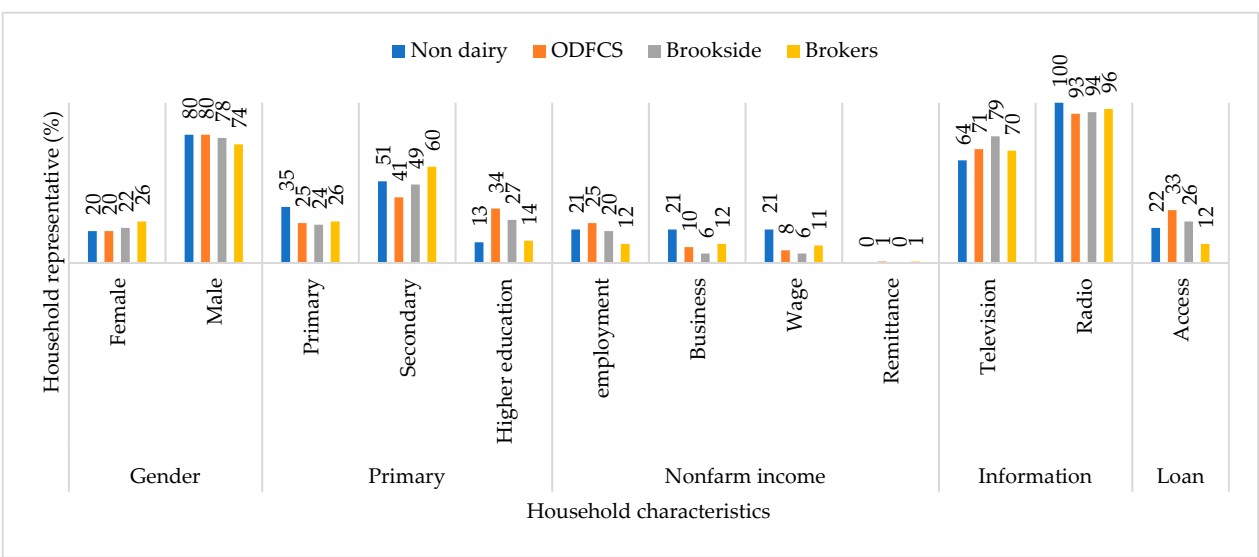

**Figure 1.** Household's representation by gender, education, non-farm income, and loan procurement.

### 3.2. Smallholders' Market Orientation and Inclination to Specific Outlet

From the descriptive results, it is apparent that not every household in Olenguruone engages in commercial dairy production, even though as the data show, at least 90% keep cattle and 83% of the non-dairy households own dairy cattle. It is indicated that farmers in non-participant group have relatively less production resources, given that they own the smallest farm plot sizes and the least livestock (dairy or non-dairy). Their annual income is aligned with their resource scarcity situation. During interviews, the cooperative extension officers added that some in this group—those keeping livestock—lack a genuine interest to engage in commercial dairy farming as an economic activity. They attributed the behavior to local culture. In a Kalenjin community, every household is expected to own dairy cattle. Furthermore, milk is an important part of the local diet, especially for young children in a household. For these households, having a cow simply secures milk for them. Their lack of interest in commercial dairy farming is exemplified by their poor livestock husbandry and not being attentive to animal dietary needs. They tend to they send the livestock to open fields and the nearby Mau forest and leave them there for days, an environment significantly lacking in nutritious animal feeds (Interviews). Drawing on these deductions, it appears that participation in commercial dairy farming seems to depend on households farming characteristics. Those identified in the present case include the level of access to production resources—the relatively well-off are engaged and have knowledge—for some, dairy farming is not considered an important economic activity beside supplying milk to the household.

Indeed, the smallholders' marking options appear to be influenced by individual households' socioeconomic attributes. A multinomial logistic regression was conducted to assess which of the attributes were associated with which marketing arrangement. Table 2 shows the results of the regression model's estimation with marketing options as the three-level dependent variable ('0' Brookside, '1' ODFCS, '3' brokers) and selective socioeconomic attributes as dependent variables—farm plot size, dairy cattle ownership, and annual income. To ensure consistency in the present model, we only incorporated the three groups engaged in commercial dairy production. Based on the results ($\chi$ = 21.95, $p$ = 0.001), there is enough information in the model to explain smallholders' market inclination. The results indicate that the smallholders' marketing outlet is significantly explained by the number of dairy cattle owned by a household. According to the estimation, an increase in dairy TLUs (milk producing cattle) by a single unit, decreases the likelihood of a household preferring ODFCS to Brookside as the market outlet by a factor of 0.76. A similar trend applies to brokers. An increase in a household's dairy cattle one unit

decreases the preference of brokers to Brookside by a factor of 0.90. Based on model's results, production capacity is the only determinant for where farmers sell their milk. An explanation to the results and information on other non-farm factors influencing marketing options are provided below.

**Table 2.** Multinomial logistic regression estimating households' dairy market inclination.

| Variables | Coef. | St.Err. | *t*-Value | *p*-Value | [95% Conf | Interval] | Sig |
|---|---|---|---|---|---|---|---|
| **Brookside** | - | . | . | . | . | . | |
| **ODFCS** | | | | | | | |
| Farm plot size (acres) | 0.39 | 0.242 | 1.61 | 0.107 | −0.084 | 0.864 | |
| Dairy TLUs | −0.763 | 0.334 | −2.28 | 0.022 | −1.418 | −0.108 | ** |
| Annual Income (KES) | −0.157 | 0.205 | −0.76 | 0.445 | −0.56 | 0.246 | |
| **Brokers** | | | | | | | |
| Farm plot size (acres) | −0.442 | 0.3 | −1.47 | 0.141 | −1.03 | 0.146 | |
| Dairy TLUs | −0.902 | 0.457 | −1.97 | 0.048 | −1.798 | −0.006 | ** |
| Annual Income (KES) | 0.009 | 0.263 | 0.03 | 0.973 | −0.506 | 0.524 | |
| Mean dependent var | | 0.972 | | SD dependent var | | 0.604 | |
| Pseudo r-squared | | 0.048 | | Number of obs | | 250.000 | |
| Chi-square | | 21.951 | | Prob > chi2 | | 0.001 | |
| Akaike crit. (AIC) | | 447.505 | | Bayesian crit. (BIC) | | 475.677 | |

** $p < 0.05$.

Although the model shows that farm plot sizes and annual income do not seem to directly explain smallholders' marketing outlet (Table 2), they certainly do contribute to the households' dairy production capacity. The descriptive results indicated that smallholder households selling their milk to Brookside constitute those most well-off in terms of production assets, having the largest farm plot sizes and highest annual income (Table 1). They also keep the most livestock (largest average TLUs). It is, therefore, reasonable that smallholders inclined to sell to Brookside as a market option have the largest dairy production capacity in the community, which is made possible by their above average assets.

Additional information from the survey data and interviews with farmers and key informants add depth to the smallholders' interest in Brookside as a marketing choice. The trust issue is brought up as among the determinants. Smallholders working with Brookside said they do not trust the cooperative, quoting management problems, inefficiency, and a lack of transparency. The cooperative management is accused by its members of deducting part of the income from daily milk supply. According to the management, the deduction is meant to be part of shareholding and a yearly bonus, yet none of the members have received the payment despite maturation period and regular follow ups. Furthermore, no explanation has been given by the cooperative leadership. The second issue involves the prices of milk for the different market outlets. Compared to ODFCS, Brookside and brokers offer better prices. The survey data shows that 54% of the smallholders selling their milk to Brookside were partly attracted to the outlet by their better prices (KES 38/litre) compared to cooperative's (KES 32/litre). In a smallholder community, KES 6/litre difference is a substantial amount.

Despite being regarded as unreliable and inconsistent, particularly by farmers working with Brookside, brokers seem to serve a proportion of farmers in the community. What motivates farmers who sell their milk to brokers? Firstly, and perhaps the most important reason is that they pay in cash at the farm gate, an option that neither Brookside nor ODFCS provides. These two pay their farmers on a monthly basis. For smallholders in this category, access to regular cash to meet their daily needs is a challenge, thus they turn to milk income (smallholder interviews). Indeed, drawing from the descriptive results (Table 1), households in this category are least represented among those with non-farm income. Overall, only 29% of households selling milk to brokers have a non-farm income

source. The second reason farmers are attracted to brokers is their better price. As indicated by 80% of all their farmers, brokers offer the highest price of the three market outlets (KES 45/litre). Indeed, cash in hand and good prices are an extra appeal to the poorest dairy farmers in the community.

Given the complaints of less pay and trust issue in the cooperative approach, the question remains: what makes smallholder in Olenguruone work with the ODFCS? The survey responses to this question indicated that ODFCS provisions to their members is the main reason that it attracts its farmers. Members of ODFCS can obtain subsidized agricultural inputs and services, including animal feeds, fertilizers, veterinary (artificial insemination) and extension services, and financial advances. They are permitted to make their payments for the advances through a check off system, which they find highly convenient (FGD). These claims align with statements made by the extension officers, who maintained that farmers in the cooperative tend to struggle with raising enough funds for farming input and other related needs, thus the possibility of borrowing from the cooperative becomes a motivation for membership. Indeed, this is among the objective of a cooperative set-up: to support smallholders to increase the quality of their milk production and productivity as well as improve the quality of produce. In light of this, establishment ODFCS seems vital for the Olenguruone dairy landscape—linking local smallholders to farming inputs so they can increase their production capacity and access to formal dairy market. While in this respect ODFCS close a vital gap in dairy farming needs in Olenguruone, the data members milk supply shows a rather uninspiring outcome. Supplying an average of 3.8 litres a day, smallholder households working with ODFCS amount to an average of 114 litres per month, which is equivalent to KES 3648 (inclusive of costs of production). Figure 2 shows the average milk supply to ODFCS in the months of October 2018 through April 2019. These low returns suggest that these farmers production capacity and the support they get from the cooperative does not amount to much. As a result, it seems the cooperative does not seem to substantially improve local livelihoods nor does it enable economies of scale, a situation which can be expected if it attracted farmers with relatively higher volumes that the current membership.

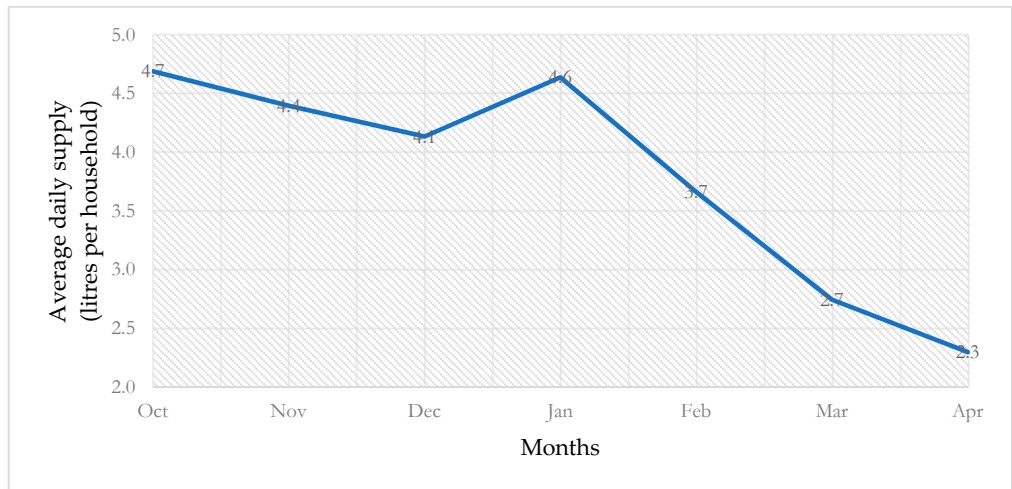

**Figure 2.** Farmers' average monthly milk supply to ODFCS.

## 4. Discussion

Generally, it is widely acknowledged among policymakers and development practitioners that commercial orientation has benefits for smallholders, especially where farmers' organizations—cooperatives—are adopted to exploit economies of scale considered impossible for the individuals [72,73]. Hence, the continued push for policies and interventions that promote smallholders' participation in commercially viable value chains through farmers' organizations such as ODFCS. ODFCS is presented as an ideal opportunity that

smallholders in Olenguruone should be keen to exploit. In doing so, however, there is no attention accorded to '[ . . . ] existing lifeworlds of the individuals and social groups affected', specifically the existing marketing opportunities in the community and the fact that some smallholders choose not to pursue dairy farming due to limited resources [10]. Simply put, the interventions promoting ODFCS did not conduct actor-oriented analyses which helps map local '[ . . . ] everyday life experiences and perceptions of the individuals and groups concerned' (ibid).

From the present research findings, it is evident smallholders' integration into a cooperative marketing opportunity, irrespective of external support, is dictated by individual smallholders' socioeconomic attributes, among other factors. This implies that the smallholders (actors) participation in the cooperative marketing arrangement or lack-thereof, is dictated their socio-economic characteristics, which essentially inform their agency on where to sell their milk. We demonstrate that the differences in the households' socioeconomic attributes ranging from farm plot sizes, family size, livestock holding, income, and income sources, to household head age and level of education, farming risk perception, and cultural factors, determine whether a household can participate in commercial dairy production and marketing arrangement.

The findings demonstrate that even in the context of external marketing support, a proportion of households in the community are still unable or unwilling to exploit the opportunity presented to them. We learn that low annual income and lack of (adequate) production assets—land and livestock—are associated with farmers' inability to participate in commercial daily farming. These two are among the primary factors that determine farmer's dairy production capacity. Hence, they should have been considered in the design and implementation of the intervention. Not doing so limits the scope of the intervention and its effectiveness. The findings are consistent with various recent studies on the 'inclusive' agribusiness interventions in smallholder communities which shows that despite external production and marketing support, a considerable proportion of smallholders are unable to exploit such opportunities due to their limited resources [74–77]. This calls for rethinking ways to support smallholders' commercialization in order to enhance their livelihood. An initial point would be acknowledging that not everyone in a smallholder community has the resources and the capacity to be involved in commercial farming. Besides the claim that farmers in the non-dairy group only keep livestock as part of local culture or are not keen to engage in commercial livestock farming, it is also evident that they comprise the households with least production assets and annual income. For this group of farmers, pursuing commercial farming agenda would be an unrealistic strategy to improve their livelihood. Alternative strategies such as promotion of off-farm economic activities and social protection programs are needed to reach them and prevent further marginalization.

The promotion of commercial farming through ODFCS in Olenguruone is considered as a way to counter the shrinking farm sizes and improving the local dairy industry. Given that a notable proportion of smallholders is not engaged in commercial dairy farming, and the monthly returns for ODFCS members, we are confident to say that this objective has only been partially met. Furthermore, we see a divergence in local marketing arrangements as a result of different needs and preferences by different smallholder households. The findings demonstrate that individual households' dairy production capacity (Dairy TLUs), financial arrangements (for instance, price of milk and cash in hand), and the trust in the outlet determine households' market choices. Several other studies share similar outcomes, that smallholders' socioeconomic attributes—farm plot size, price of the produce, size of the dairy cattle holding, among others—and market perceptions influence the marketing channel through which individual households sell their produce [20,22,23,25,69,70,78]. Mburu et al. [25], for instance, find that an increase in milk price, size of dairy cattle holding, and farm size have a negative influence on smallholders marketing their produce with a cooperative. According to Ollila, Nilsson [78] as cited in Cechin, Bijman [79], 'farms with larger production capacity are likely to be less dependent on the cooperative for

market access and, therefore, less willing to be loyal when they see short-term economic benefits outside'. This corroborates the present research findings; specifically with respect to farmers marketing their milk with Brookside. These findings suggest that the cooperative is characteristically an option for farmers in the middle in terms of production capacity (resource level) in the community—not for the poorest nor the most well-off.

The problem of mistrust towards ODFCS is not a new issue affecting cooperatives membership in Kenya. As reported by Wanyama [80], incidences of mismanagement and corruption within cooperatives date back to the 1990s following the 1997 Cooperative Societies Act that 'empowered the members to be responsible for the running of their own cooperatives, through elected management committees'. Notwithstanding the Act amendment in 2004 that gave the state the mandate to regulate cooperatives in the country, many related challenges persist, including poor financial management, leadership and governance, and political interference [80–82]. It seems these issues remain a concern even in the context of donor funded cooperatives such as ODFCS where some local smallholders highlight a lack of transparency with respect to financial management (deductions). The mistrust towards ODFCS highlights the risk perception some farmers have towards this marketing arrangement. The same applies to the finding that some farmers find brokers unreliable and inconsistent. The farmers' perception of the market, therefore, brings an additional dimension to diversity in the community that has an influence on the market channel to which individual households' lean.

Irrespective of the outlined motivations against ODFCS membership, our findings demonstrate that to a certain extent, the cooperative serves an important role in the local commercial dairy industry. Through collective marketing, smallholders working with ODFCS can collectively access subsidized input and services to enhance their production, at a convenient repayment arrangement: the checkoff system. Such an opportunity is not available to farmers supplying milk to Brookside or brokers should they need it. In addition, ODFCS's linkage to Happy Cow, ensures farmers unable to supply milk to a formal channel such as Brookside can still participate and benefit from the formal dairy market through the cooperative arrangement. Nevertheless, given how low the average milk volumes supplied by ODFCS members, the impact of the cooperative to local livelihoods remains highly limited. The little local contribution of ODFCS in Olenguruone implies that the cooperative members may adversely incorporated in the dairy commercial value chain [83,84]. Combined with the fact that the ODFCS does not reach the most resource poor, the approach and/or the level of support to smallholders may need rethinking.

Cooperatives thrive on volumes, which is the reason they are promoted as a collective effort necessary among smallholders to achieve economies of scale [85]. By failing in attracting those with relatively higher production capacity, ODFCS is unlikely to regularly collect volumes that are critical for growth. At the same time, it is crucial to acknowledge that one size does not fit all. The different marketing channels seem to cater for the diversity of marketing needs and priorities in Olenguruone's smallholder community. It would be unrealistic to expect that these needs and priorities can be met by one marketing channel. In this context, future interventions should consider multiple options to promote.

## 5. Conclusions

The present study set out to explore the commercial farming orientation and inclination towards specific marketing arrangements based on individual smallholders' socioeconomic attributes. The findings demonstrate that even in the context of relatively close rural smallholder communities, there is a considerable level of heterogeneity in the households' socioeconomic characteristics that dictate farming systems and marketing arrangements. Indeed, individual smallholders' have agency towards participation in market-oriented production and specific marketing arrangements as determined by their unique farming situations—resources, payment systems, attitudes/risk perception, among others. Therefore, an analysis of individual smallholders' context vis-à-vis engagement in commercial farming and how to best meet their marketing needs in a manner that ad-

vances their livelihoods is critical. Accordingly, those promoting the farmers' organizations such as cooperatives ought to acknowledge the presence of important diversity in local marketing needs and priorities emerging from the difference in socioeconomic attributes that characterize individual smallholder households.

**Author Contributions:** All authors made a significant contribution to the present manuscript preparation. J.W. was involved in conceptualization, data collection, data analyses and drafting the article. E.M. and A.C.M.v.W. were supervisors and collaborators of this project, aided in reviewing the manuscript. All authors have read and agreed to the published version of the manuscript.

**Funding:** This work was financed by the Netherlands Organization for Scientific Research—NWO-WOTRO, (Grant number W 08.250.206). It is a part of the Follow the Food research project by Utrecht University and partners that assesses the contribution of inclusive agribusiness to local food security in Africa.

**Informed Consent Statement:** An informed concept was obtained from all participants of this study prior to their involvement.

**Data Availability Statement:** Data are available upon request from the corresponding author, [J.W.].

**Conflicts of Interest:** The authors declare no conflict of interest.

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
