# Peer review of "Recognizing Determinants to Smallholders’ Market Orientation and Marketing Arrangements: Building on a Case of Dairy Farming in Rural Kenya"

_land, doi:10.3390/land10060572_

Round 1
Reviewer 1 Report
Abstract: the abstract lacks important aspects such as aim, methods to gather data and the clear presentation of findings.
Keywords: the key words already covered in the heading should not be should be replaced.
Introduction: The authors try to give background information on market orientation in smallholder farming and marketing strategies. However, clear definition of the “Smallholder” or “small-scale” agriculture should be provided in this context. The text under the heading “Kenya dairy industry” should be summarized and form part of the Introduction. The authors should also provide a clear aim of this study in the concluding paragraph of the introduction.
Materials and methods: A heading “Data analysis” should be created to cover the text presented on the last paragraph in the Materials and Methods section. The authors pointed out the STATA (Version 13) statistical programme will be use to analyses data for frequency and summary table, and t-tests. It is imperative to indicate variables to be analyzed through the above-mentioned statistical tools.
Findings: The text under this heading presents the results and discussion. It would be appreciated if the authors consider either 1) replacing the heading “Findings” with “Results and discussion” OR 2) present the Results and discussion on separate headings. Text presented on the heading “ 4.3. Understanding smallholders’ market orientation and inclinations” provides interrogations of the current findings with previous studies. Therefore, this should form part of the discussion. The last paragraph in this section (page 12) should be covered by the heading “ Conclusion”.
Table 1: In the first column of this table, a heading covering variables such as age, family size, farm plot size, etc should be added. A unit of area in the metric system (e.g. ha or acre) should be indicated to specify plot size. The same applies to the rest of the variables in this column, if applicable.
Figure 1: This figure does not provide full explanation of the findings. I suggest that a heading for both X and Y-axis should be inserted.
Table 2: A unit of area in the metric system (e.g ha or acre) should be indicated to specify plot size. The same applies to the rest of the variables in this column, if applicable.
Figure 2: This figure also does not provide full explanation of the findings. This may be improved by I inserting heading for both X and Y-axis.
References: Please use numeric citation to number each of your sources in the reference list . This may disqualify alphabetic listing of your references.
Author Response
Reviewer 1 comments and response
Comments and Suggestions for Authors
Abstract: the abstract lacks important aspects such as aim, methods to gather data and the clear presentation of findings.
Keywords: the key words already covered in the heading should not be should be replaced.
- Has been addressed
Introduction: The authors try to give background information on market orientation in smallholder farming and marketing strategies. However, clear definition of the “Smallholder” or “small-scale” agriculture should be provided in this context. The text under the heading “Kenya dairy industry” should be summarized and form part of the Introduction. The authors should also provide a clear aim of this study in the concluding paragraph of the introduction.
- Definition of smallholder(s) has been included in the past paragraph in the introduction: Line 28-30
- A brief summary and introduction to the text under “Kenya dairy industry” has been included in from Line 109 to 112
- The aim of the study is presented in from Line 45-49; This paper explores …
Materials and methods: A heading “Data analysis” should be created to cover the text presented on the last paragraph in the Materials and Methods section. The authors pointed out the STATA (Version 13) statistical programme will be use to analyses data for frequency and summary table, and t-tests. It is imperative to indicate variables to be analyzed through the above-mentioned statistical tools.
- The sub-heading – Empirical methods, we implied data analysis; I have renamed the sub-section data analyses
- The text leading up to as well as the text following the part sentence mentioning STATA (version13) provide extensive information on the variables used for the analyses
Findings: The text under this heading presents the results and discussion. It would be appreciated if the authors consider either 1) replacing the heading “Findings” with “Results and discussion” OR 2) present the Results and discussion on separate headings. Text presented on the heading “ 4.3. Understanding smallholders’ market orientation and inclinations” provides interrogations of the current findings with previous studies. Therefore, this should form part of the discussion. The last paragraph in this section (page 12) should be covered by the heading “ Conclusion”.
- These issues have been addressed
Table 1: In the first column of this table, a heading covering variables such as age, family size, farm plot size, etc should be added. A unit of area in the metric system (e.g. ha or acre) should be indicated to specify plot size. The same applies to the rest of the variables in this column, if applicable.
Figure 1: This figure does not provide full explanation of the findings. I suggest that a heading for both X and Y-axis should be inserted.
- Addressed
Table 2: A unit of area in the metric system (e.g ha or acre) should be indicated to specify plot size. The same applies to the rest of the variables in this column, if applicable.
- Addressed
Figure 2: This figure also does not provide full explanation of the findings. This may be improved by I inserting heading for both X and Y-axis.
- Addressed
References: Please use numeric citation to number each of your sources in the reference list . This may disqualify alphabetic listing of your references.
- Addressed
We are highly grateful for your comments on our manuscript and we hope to have adequately integrated them and/or provided a satisfactory response.

Reviewer 2 Report
- The manuscript requires major editing for language; too many instances of missing words, subject-verb disagreement, misplacement of verbs and pronouns, etc.
- Regarding objective; the authors state on lines 15-16 as follows: “the present research investigates the factors that determines smallholders’ commercial farming orientation and marketing options”. Then on lines 109 and 110, the authors state as follows: "the present research investigates the factors that determines smallholders’ commercial farming orientation and marketing options”. This is repeated on lines 254-254, as follows: “The objective of the present research was to determine individual farmer to engage in commercial dairy farming and marketing option”. Then on lines 549-550, the authors also state as follows: “the present paper set out to explore the commercial farming orientation and inclination towards specific marketing channels based on individual smallholders’ socioeconomic attributes”. Authors should avoid restating the objective of the paper severally in the paper. They should streamline it and state it only once and maybe rehashed it in the concluding sections of the paper.
- The paper uses a relevant Actor-oriented approach. However, in the analysis sections, not much of this was brought to bear.
- It appears the manuscript does not have a conclusion. On lines 113/114, the authors state that “the findings are discussed, and the conclusions are drawn in section five”. However, there is no section 5 and no section on conclusions in the manuscript at all.
Author Response
Reviewer 2 comments and response
Comments and Suggestions for Authors
1. The manuscript requires major editing for language; too many instances of missing words, subject-verb disagreement, misplacement of verbs and pronouns, etc.
- The language editing has been done
2. Regarding objective; the authors state on lines 15-16 as follows: “the present research investigates the factors that determines smallholders’ commercial farming orientation and marketing options”. Then on lines 109 and 110, the authors state as follows: "the present research investigates the factors that determines smallholders’ commercial farming orientation and marketing options”. This is repeated on lines 254-254, as follows: “The objective of the present research was to determine individual farmer to engage in commercial dairy farming and marketing option”. Then on lines 549-550, the authors also state as follows: “the present paper set out to explore the commercial farming orientation and inclination towards specific marketing channels based on individual smallholders’ socioeconomic attributes”. Authors should avoid restating the objective of the paper severally in the paper. They should streamline it and state it only once and maybe rehashed it in the concluding sections of the paper.
- All these issues have been addressed
3. The paper uses a relevant Actor-oriented approach. However, in the analysis sections, not much of this was brought to bear.
- Indeed, we had not clearly shown how Actor-oriented approach informed the rest of the paper after the introduction. We have now clearly integrated it in discussing the findings and briefly in the conclusions. Please see Lines: 455-468 and 556-558
4. It appears the manuscript does not have a conclusion. On lines 113/114, the authors state that “the findings are discussed, and the conclusions are drawn in section five”. However, there is no section 5 and no section on conclusions in the manuscript at all.
- This issue has been addressed
We are highly grateful for your comments on our manuscript, and we hope to have adequately integrated them and/or provided a satisfactory response.

Round 2
Reviewer 1 Report
Introduction: this section still require some further amendments. For examples, on line 112-115 the authors included the text indicating the outline of the manuscript ( Section three will provide information on the case and study area, methods of data collection and analyses. In section four, the main findings are presented, which are then discussed in section five. Conclusions are drawn in section six). This should be deleted.
References cited: There are still some few references cited not in line with the journal style of referencing.
See the rest of comments in the attached manuscript
Kenya dairy industry: This section should form part of general introduction as it outlines the overview of Kenyan dairy industry. This also forms the basis of problem statement for the current study. Therefore the aim of the study should be presented immediately after this.
Findings Figure 1- The title on the Y-axis should be improved as follows: Household representative (%). The text in line 335-340 should be aligned. The title for Y-axis on Figure 2 should be rephrased as "Average daily supply (Litre per household)".

Author Response
Dear Reviewer,
Thank you, again, for your very helpful comments and suggestions. Please find my response below.
Comments and Suggestions for Authors
Introduction: this section still require some further amendments. For examples, on line 112-115 the authors included the text indicating the outline of the manuscript ( Section three will provide information on the case and study area, methods of data collection and analyses. In section four, the main findings are presented, which are then discussed in section five. Conclusions are drawn in section six). This should be deleted.
References cited: There are still some few references cited not in line with the journal style of referencing.
See the rest of comments in the attached manuscript
Kenya dairy industry: This section should form part of general introduction as it outlines the overview of Kenyan dairy industry. This also forms the basis of problem statement for the current study. Therefore the aim of the study should be presented immediately after this.
Findings Figure 1- The title on the Y-axis should be improved as follows: Household representative (%). The text in line 335-340 should be aligned. The title for Y-axis on Figure 2 should be rephrased as "Average daily supply (Litre per household)".
All these suggestions have been integrated:
- I deleted the text that you suggested I do.
- Fixed the references to fit the journal style
- Made the text on Kenya’s dairy industry part of the introduction (as a sub-section of introduction), and toward the end of it, included new text on the aim of the study.
- Fixed the titles on the figures and aligned text for one of the figures as suggested.
- Proof-read the text again to fix minor language mistakes and to improve readability as well as argumentation